# Ni/cerium Molybdenum Oxide Hydrate Microflakes Composite Coatings Electrodeposited from Choline Chloride: Ethylene Glycol Deep Eutectic Solvent

**DOI:** 10.3390/ma13040924

**Published:** 2020-02-19

**Authors:** Juliusz Winiarski, Anna Niciejewska, Jacek Ryl, Kazimierz Darowicki, Sylwia Baśladyńska, Katarzyna Winiarska, Bogdan Szczygieł

**Affiliations:** 1Faculty of Chemistry, Department of Advanced Material Technologies, Wrocław University of Science and Technology, 50-370 Wrocław, Poland; anna.niciejewska@onet.pl (A.N.); sylwia.basladynska@pwr.edu.pl (S.B.); bogdan.szczygiel@pwr.edu.pl (B.S.); 2Faculty of Chemistry, Department of Electrochemistry, Corrosion and Materials Engineering, Gdansk University of Technology, 80-233 Gdańsk, Poland; jacek.ryl@pg.edu.pl (J.R.); kazimierz.darowicki@pg.edu.pl (K.D.); 3Faculty of Chemistry; Department of Analytical Chemistry and Chemical Metallurgy, Wrocław University of Science and Technology, 50-370 Wrocław, Poland; katarzyna.winiarska@pwr.edu.pl

**Keywords:** metal coatings, nickel, composite coatings, electrodeposition, XPS, polarization, electrochemical impedance spectroscopy (EIS)

## Abstract

Cerium molybdenum oxide hydrate microflakes are codeposited with nickel from a deep eutectic solvent-based bath. During seven days of exposure in 0.05 M NaCl solution, the corrosion resistance of composite coating (Ni/CeMoOxide) is slightly reduced, due to the existence of some microcracks caused by large microflakes. Multielemental analysis of the solution, in which coatings are exposed and the qualitative changes in the surface chemistry (XPS) show selective etching molybdenum from microflakes. The amount of various molybdenum species within the surface of coating nearly completely disappear, due to the corrosion process. Significant amounts of Ce^3+^ compounds are removed, however the corrosion process is less selective towards the cerium, and the overall cerium chemistry remains unchanged. Initially, blank Ni coatings are covered by NiO and Ni(OH)_2_ in an atomic ratio of 1:2. After exposure, the amount of Ni(OH)_2_ increases in relation to NiO (ratio 1:3). For the composite coating, the atomic ratios of both forms of nickel vary from 1:0.8 to 1:1.3. Despite achieving lower corrosion resistance of the composite coating, the applied concept of using micro-flakes, whose skeleton is a system of Ce(III) species and active form are molybdate ions, may be interesting for applications in materials with potential self-healing properties.

## 1. Introduction

In recent years, the electrodeposition of metals from ionic liquids (ILs) has become more popular because of the many advantages, such as high solubility metals salts, high conductivity, and wide electrochemical potential windows. Ionic liquids also allow obtaining metal coatings, such as Na and Mg, which is impossible in conventional baths. ILs are non-toxic, non-flammable and can be used in a wide range of temperatures [1,2,3,4]. One of the most interesting analogues of ILs are the deep eutectic solvents (DESs). DES is a eutectic mixture of quaternary ammonium salt with hydrogen bond donor species or metal salts. DESs are relatively cheap and easy to prepare [5,6]. DES-based galvanic baths have the advantage that the metals do not passivate during deposition. DESs can be used to electrodeposition a lot of metal coatings: Ni, Cu, Zn, Cr, Sn, Co, Al, Ti, W, Ag, Mg, Se, and Pd [6]. Nickel coatings obtained from DES feature nano-crystalline morphology and lower surface roughness than Ni-coatings from conventional aqueous Watt’s baths which produce coatings with a micro-crystalline structure. Furthermore, the microhardness of Ni coatings deposited from DES is 100 HV (Vickers hardness) higher than coatings from aqueous solutions [7]. A used eutectic solvent has a significant impact on the morphology of nickel coatings (size of Ni crystallites) [8]. Moreover, from baths based on eutectic solvents it is possible to obtain Ni coatings on an aluminum substrate. Obtaining bright Ni coatings requires the use of brighteners, which differ from those used in aqueous plating baths [9,10]. The important influence on the coating’s deposition from DESs is the water content in a plating bath. By changing the water content, one can control the potential for Ni deposition. Potential Ni impacts the self-limiting growth crystallite and passivation effect [11]. In addition to the above advantages, DESs liquids enable the electrodeposition of nickel alloy coatings: Zn-Ni [12], Ni-Co-Sn [13], Fe-Ni [14], Ni-Co [15], Ni-Mo [16], and nickel composite coatings. Ni-carbon nanotubes composite coatings [17,18] have much better tribological properties than pure Ni [18]. Furthermore, Ni-PTFE (poly(tetrafluoroethylene)) coatings show better wear resistance than nickel coatings and have hydrophobic properties [19]. The addition of SiC to the nickel matrix causes the increase of microhardness [20] and corrosion resistance [21]. Ni-SiO_2_ coatings also have better microhardness and wear resistance than Ni coatings [22]. Nickel coatings with a TiO_2_ dispersive phase show good electrocatalytic activity [23]. However, co-deposition with micro- or nano-particles of compounds of chemically active elements, e.g. cerium or molybdenum, may affect the protective properties of the composite coatings. Dong et al. prepared cerium molybdate particles with different morphologies, such as the bundle-like, flower-like, and microspheric structures [24]. The nanostructure of Ce-Mo shows excellent adsorption and catalytic properties [24]. Amino acids, for example: glycine, lysine, serine, and glutaminate used at synthesis of cerium molybdate have influence on its morphology [25]. Sajad et al. proved that by using lysine in synthesis one can obtain a pure product (cerium molybdate) which is not contaminated with cerium molybdate oxides [26]. Nanocapsules are also made up of shells from cerium molybdate, while the core was a corrosion inhibitor [27,28] or hollow nanospheres have an antibacterial effect on *Escherichia coli* in the absence of light [29]. Hence their potential application in medicine or the anti-corrosion industry [29]. Patel et al. developed a method of fabrication of cerium zinc molybdate nanopigment with anticorrosive properties [30]. Cerium molybdate nanowires were also added to the coatings obtained by the sol-gel method [31]. Therefore, cerium molybdate is very popular as a corrosion inhibitor in protective coatings for aluminum alloys [32] and magnesium [33,34]. Bhanvase et al. have investigated that 5% content of cerium zinc molybdate nanocontainers, significantly increases the corrosion resistance [35].

The use of cerium molybdenum oxide, proposed in this work, is novel in electrodeposition of nickel composite coatings. Ce and Mo compounds embedded in the coating can modify its protective properties by giving it potential self-healing properties (through the selective release of cerium and molybdenum ions). Moreover, this work uses a deep eutectic solvent based on choline chloride and ethylene glycol as an excellent environment for the co-deposition of microparticles. As a result, deposition is carried out in a stable suspension bath. Microflakes of fairly large size are intentionally used to make it easier to capture the effect of their interaction with a metallic nickel matrix and the corrosive environment. Emphasis is placed on understanding qualitative and quantitative changes in surface chemical composition—X-ray photoelectron spectroscopy (XPS), caused by the presence of microflakes, on the one hand, and the exposure to a corrosive environment, on the other. The corrosion resistance of a newly obtained material is monitored in-situ by electrochemical impedance spectroscopy (EIS) and basic *dc* (direct current) polarization methods. The morphology, structure, and topography of the coatings are investigated by scanning electron microscopy (SEM), X-ray diffraction (XRD), and contact profilometry. 

## 2. Materials and Methods

Cerium molybdenum oxide microflakes were synthesized at low temperature from Ce(NO_3_)_3_ and Na_2_MoO_4_ precursors through the precipitation method. First, the two precursors were prepared separately by the dissolution of Na_2_MoO_4_ 2H_2_O and Ce(NO_3_)_3_ 6H_2_O in demineralized water in the molar ratio of 2:1, respectively. The solutions were next cooled down to 8 °C overnight in the fridge. Na_2_MoO_4_ solution was instilled slowly (60 mL min^−1^) into Ce(NO_3_)_3_ solution under continuous mechanical stirring (200 rpm), which resulted in the formation of a light yellow suspension. After 1 h mixing at 15 °C, the precipitate was filtered under the reduced pressure on a Buchner funnel. The grey-yellowish precipitate was washed several times with demineralized water and then dried in a vacuum dryer.

The plating bath for nickel electrodeposition (100 mL plating volume) was prepared by mixing choline chloride and ethylene glycol in a 1:2 molar ratio and addition of 1 mol dm^−3^ NiCl_2_ 6H_2_O at 70 °C. All these reagents were mechanically stirred until a homogeneous green liquid was obtained. Cerium molybdenum oxide microflakes were then added (2 g dm^−3^) and the bath was homogenized for 30 min (UP50H – Hielscher Compact Lab Homogenizer).

For the electrodeposition, copper disks (1.5 cm diameter, 0.1 cm thick) were used as cathodes. They were polished on 600–1200 grade abrasive paper. Then, the samples were degreased in an ultrasonic cleaner in methanol, acid etched (10 vol.% H_2_SO_4_), rinsed in demineralized water and again in methanol. Electroplating was carried out in a thermostatic electrolyzer consisting of two anodes (pure nickel) and a cathode placed between them. The process was carried out at a current density 6 mA cm^−2^, at 70 °C, for 1 h. After this time, the coatings were twice rinsed in demineralized water and methanol. At the end, the coatings were dried and stored in a vacuum desiccator until required.

Surface morphology was analyzed by Quanta 250 (FEI) scanning electron microscope equipped with an Octane Elect Plus SDD microanalyzer (operating at 25 kV, and at 10^−4^ Pa pressure).

The crystal structure and phase composition of cerium molybdenum oxide microflakes and blank and composite nickel coatings were analyzed using powder X-ray diffraction. The XRD diffractograms were recorded on a D 5000 (Siemens) diffractometer with CuKα radiation (λ = 0.15409 nm) at room temperature in the 2*θ* range: 10–60° for blank and composite Ni coatings, and 5–60° for cerium molybdenum oxide microflakes. Phase identification was carried out by comparing the experimental patterns with the reference patterns collected in the Powder Diffraction Files database (International Centre for Diffraction Data PDF-2 base).

The X-ray Photoelectron Spectroscopy (XPS) measurements were performed using Escalab 250Xi (Thermo Fisher Scientific, United Kingdom). Al Kα monochromatic X-ray source with spot diameter of 250 μm was used. The set up pass energy was 20 eV. Charge compensation was controlled through low-energy Ar^+^ ions emission by means of a flood gun, with the final calibration made using Ni2p3 metallic peak component at 852.6 eV. The deconvolution procedure was performed using Avantage software (Thermo Fisher Scientific, Waltham, USA).

The chemical analysis of corrosive solutions was performed by ICP-OES (Inductively Coupled Plasma-Optical Emission Spectrometry) and ICP-MS (Inductively Coupled Plasma-Mass Spectrometry) technique. The Ni concentration was determined using ICP-OES method (Vista MPX, Varian, Australia). Cu, Ce, Mo, and also Ni content was determined using ICP-MS method (XSeries2, Thermo Fisher Scientific, USA). The analyses were performed in the Chemical Laboratory of Multielemental Analysis, Wrocław University of Science and Technology, Poland, accredited by the Polish Centre for Accreditation (AB 696). The laboratory has a measurement procedure of determination of metals in water in its scope of accreditation. The procedure is based on EN-ISO 11885:2009, PN-EN ISO 17294-1:2007, and PN-EN ISO 17294-2:2006.

Corrosion measurements were carried out at 25 °C in 0.05 mol dm^−3^ solution of NaCl in Autolab 400 mL corrosion cell using a Reference 1010E (Gamry, Warminster, USA) potentiostat/galvanostat/ZRA (zero resistance ammeter). The geometric area of the working electrode exposed to the solution was 1 cm^2^. A stainless steel rod (geometric area 5 cm^2^) and a saturated Ag|AgCl / 3M KCl electrode (Metrohm) mounted in a Luggin capillary were used as the counter and reference electrodes, respectively. Potentiodynamic polarization curves were recorded after 168 hours exposure starting from −0.1 V to +0.5 V vs. open circuit potential (*E*_OC_) at a scan rate of 0.166 mV s^−1^. The polarization resistance (LPR technique) was measured by polarization of the sample starting from −10 mV to +10 mV vs. *E*_OC_ at a scan rate of 1 mV s^−1^. Impedance spectra for electrochemical impedance spectroscopy (EIS) were recorded at *E*_OC_ (potentiostatic mode) with a resolution of 10 pts/dec., in a frequency range from 100 kHz to 0.001 Hz and at a 10 mV signal amplitude. Equivalent circuit modeling, graphing, and analysis of impedance data was performed using ZView^®^ software (Scribner Associates, version 3.5g). The deposition process was studied by cyclic voltammetry (CV) technique using Autolab RDE-2 electrode with a Pt tip (3 mm diameter) in a electrochemical vessel (20–90 mL, Metrohm) with thermostat jacket, Ag and Pt wires as the reference and counter electrodes, in the range of potentials from *E*_OC_ to −1.5, then 1.5 and ending at *E*_OC_ with a scan rate of 20 mV s^−1^. Pt tip instead of Cu was chosen to avoid the influence of any copper ions on the current-potential characteristics. 

## 3. Results and Discussion

### 3.1. Morphology, Topography, and Phase Structure of Blank and Composite Ni Coatings

After electrodeposition, both blank and composite nickel coatings with a mean thickness of 10 µm were subjected to SEM and XRD analysis. The surface of the Ni coating consisted of spheroidal particles that form randomly arranged larger agglomerates (Figure 1a). The surface of the Ni/CeMoOxide coating, like the Ni coating, consisted of spherical particles, which were noticeably smaller in this case. In addition, there were randomly embedded microflakes of CeMoOxide within the surface. Single cracks were visible, mostly spreading from the places where the microflakes are built-in (Figure 1b). Additional topographic measurements of the surface of both coatings confirmed the increase in roughness of the composite coating, expressed by the R_a_ parameter equaled to 122.3 ± 17.7 nm (related to 92.6 ± 10.1 nm for blank Ni coating). For comparison, Figure 1c shows the photomicrograph (recorded in a secondary electron - SE mode of SEM) of synthesized cerium molybdenum oxide hydrate powder. The plates are overgrown and form agglomerates with a length exceeding 20 μm, a width below 10 μm, and a thickness of 100–200 nm (Figure 1c).

Figure 2 shows the diffractograms for of cerium molybdenum oxide microflakes and blank and composite nickel coating. The thickness of the prepared Ni coatings was sufficient to perform the XRD analysis. The dominant peaks at the diffractograms are characteristic of Ni (111) and (200) crystal planes visible at 2θ 44.6° and 51.9°. The indicated reflections were well indexed to these collected in ICDD JCPDS card No. 00-004-0850 for Ni. Peaks at 2θ 43.3° and 50.4° originated from the Cu substrate (JCPDS card No. 00-004-0836). The other small peaks match well with the cerium molybdenum oxide. The diffractogram recorded for as-prepared microflakes is also shown in Figure 2. All the diffraction peaks correspond to Ce_2_(MoO_4_)_3_ 4.5H_2_O phase (JCPDS card No. 00-031-0333) without the presence of other peaks from any impurities. The strong and sharp diffraction peaks indicate that the prepared cerium molybdenum oxide microflakes were well-crystallized. High-resolution diffractograms were recorded for the blank and composite Ni coatings to estimate the average crystallite size. A copper peak from the substrate overlapped the nickel peak (111), therefore, for the proper determination of FWHM (full width at half maximum) for the Ni (111) peak, a deconvolution using a pseudo-Voigt function was performed. The average crystallite size was calculated using the Scherrer Equation (1):(1)$$D=\;\frac{K\cdot \lambda }{B\cdot cos\theta }$$
where D is the mean crystallite size in the direction perpendicular to the (hkl) plane of reflexes in nm, K is a Scherrer constant (0.9) [36], λ = 0.154 nm is the X-ray wavelength used in the measurement, B was calculated from Equation (2): (2)$$B=\;\sqrt{(\beta_{FWHM}^{2}-\;\beta_{0}^{2}})$$

(β_FWHM_ and β_0_ is the FWHM of diffraction peak at angle θ and the corrected instrumental broadening - in radian, respectively). After deconvolution, the average crystallite size was estimated at about 10.4 nm and 6.3 nm for blank Ni and composite Ni coatings respectively. The resulting D value indicates the nano-crystalline structure of the electrodeposited nickel. It can be seen that the cerium molybdenum oxide microflakes affects the structure of the coating by co-depositing and reducing the size of the crystallites.

On the basis of the observed changes in the morphology and structure of both types of coating, it is obvious that the cerium and molybdenum compound must have modified the course of cathodic processes. Therefore, cyclic voltammetry curves were recorded under deposition conditions in both (blank Ni and suspension) plating baths. The experiment was repeated twice to avoid accidental results. Representative CV curves are shown in Figure 3. Analyzing the course of the curves, it can be seen that the composite coating deposition can proceed probably with greater efficiency. This can be demonstrated by a much larger anodic peak within the potentials of Ni (deposit) oxidation (Figure 3). It is however not excluded that the larger area of that peak may be associated with, e.g., the oxidation of molybdenum species released at high temperature (70 °C) from the microflakes suspended in the plating bath. However, the cathode curves were not significantly different (Figure 3).

### 3.2. Corrosion Resistance

#### 3.2.1. *dc* Polarization

Polarization resistance (*R*_p_) as a function of exposure time in 0.05 mol dm^−3^ solution of NaCl for blank Ni and composite (Ni/CeMoOxide) coatings is presented in Figure 4. In the case of a blank Ni coating, a clear increase in *R*_p_ was observed. During exposure in a NaCl solution, *R*_p_ increased up to 536.6 kΩ cm^2^ after about 144 hours, then, its value decreased. The *R*_p_-time dependency for the Ni/CeMoOxide coating was different. An increase of *R*_p_ to above 100 kΩ cm^2^ was visible within the first hours of exposure in a chloride solution. After approximately 72 hours, the *R*_p_ value decreased to approximately 77 kΩ cm^2^, and then it showed a slight upward trend (Figure 4).

The corrosion potential (*E*_corr_) of the Ni/CeMoOxide coating increased quite rapidly in the first hours (Figure 5). After about three days it reached a constant level. The potential-time dependency for the Ni coating was different. An increase in *E*_corr_ towards more positive values was noticeable throughout the entire time studied. It is possible that such a direction of changes resulted from qualitative changes in the morphology of the layer of nickel oxidation products.

Finally, after 168 hours of exposure—after reaching a certain stability of the potential (at least for the composite coating), polarization curves for Ni and Ni/CeMoOxide coatings were registered. Their course is presented in Figure 6. The anodic branch of the Ni/CeMoOxide coating, compared to the anodic branch of the Ni coating, is characterized by higher corrosion current density values, which suggests more active anodic process on the surface of the former. A section with slowly a growing density of the corrosion current is visible on the anodic side in the course of Ni coating (Figure 6), which may suggest some protective/barrier action of a passive layer.

#### 3.2.2. Electrochemical Impedance Spectroscopy

Impedance spectra of both types of coating were recorded every 24 hours during 7 days of exposure of the samples in 0.05 mol dm^−3^ NaCl solution. They have been presented on a complex Bode plot (Figure 7) to gain better visibility of the changes of the investigated system. The first analysis of the shape of the spectra leads to the conclusion that the corrosion process of Ni coating is characteristic of “corroding coating”. The magnitude of impedance |Z| increased during exposure (Figure 7a), which may indicate increasing corrosion resistance, due to the formation of, e.g., a stable passive layer or increasing diffusion constrains (especially after 144 and 168 hours). A similar shape of impedance spectra and similar tendency (increasing) for the impedance modulus have been observed in previous work by Urcezino et al. [37] and J. Winiarski et al. [38]. 

Figure 7b presents impedance spectra for composite coating. This material behaved slightly differently, rather like “damaged coating”, due to the presence of coating incontinuities. Furthermore, not very clearly separated time constants—caused by capacitive dispersion and overlapping of time constants—were observed (Figure 7b). Therefore, for further discussion, two electric equivalent circuits (EECs) were used for the calculation of the theoretical spectra: a single time constant model for a blank Ni (Figure 8a) and a double time constant model for a composite Ni (Figure 8b).

Both models use a constant phase element (CPE). The impedance of the CPE is defined by Equation (3), where: *Y*_0_ is a time constant parameter (Ω^−1^ cm^−2^ s^α^), *ω* is the angular frequency of the AC signal and *α* is the CPE exponent.
(3)$$Z_{CPE}=Y_{0}^{-1}{\left(j\omega \right)}^{-\alpha }$$

The other elements used in the EECs presented in Figure 8a,b are: *R*_s_—the resistance of NaCl solution, *R*_ct_—the charge transfer resistance associated with nickel oxidation, *Y*_0,dl_ and *α* corresponds to a double layer capacitance (*C*_dl_). In the circuit for a composite coating (Figure 8b), the properties of a developed surface layer of corrosion product was modeled using *R*_film_ (the resistance of electrolytic solution in a porous layer) and *C*_film_ (the dielectric properties of this layer). Both EECs yielded a very good fit to the experimental data (parameter χ_2_ in order of 10^–4^–10^–5^) and low residual errors (0.2%–8%). However, it should be noted that the fitting was made after manually reducing the frequency range to ca. 10 kHz–0.08 Hz. This interference was intentional to avoid the influence of instability of the measured system, especially in the low frequency range, on the interpretation of the physical meaning of EECs elements and, finally, their values (collected in Table 1 and Table 2). 

For a blank Ni coating, the values of *R*_ct_ increased with increasing exposure time and finally exceeded 0.5 MΩ cm^2^ after 7 days (Table 1). A similar tendency was observed for the *R*_p_ determined on the basis of the LPR technique, see Figure 4. This marked increase in *R*_ct_ was accompanied by only minor changes in CPE parameters (*Y*_0_ and *α*), whose values indicate the absence of significant changes in (e.g., homogeneity) of the surface at which the corrosion process occurs. In the case of a composite coating (Table 2), it was also noted that the direction of changes of a charge transfer resistance was close to the polarization resistance values obtained from the LPR method (Figure 4). Initially, for the first 48 hours, there was a drop of *R*_ct_ from 75 to 59 kΩ cm^2^, but after 72 hours exposure, the values of *R*_ct_ increased with increasing time and finally exceeded 100 kΩ cm^2^ after 7 days (Table 2). For this coating the contribution of a resistance connected with the presence of a developed surface layer of corrosion products is not to be missed—*R*_film_ values increased from 5.6 to 8.8 kΩ cm^2^ over 7 days exposure (Table 2).

### 3.3. XPS Surface Analysis of Coatings

In order to better understand the role of microflakes in the corrosion process, both composite and blank Ni coatings were analyzed by X-ray photoelectron spectroscopy (XPS). The high-resolution XPS analyses were performed in the binding energy (BE) range of Ni2p, Cu2p, and O1s photopeaks, with the results presented in Figure 9. Supporting studies were also carried out in the C1s BE range.

The analysis performed for the blank Ni coating prior to corrosion stability tests reveal a weak signal from nickel in the Ni2p spectra (see Figure 9a), further deconvoluted using two peak doublets Ni_(B)_ and Ni_(C)_. The Ni2p_3/2_ photopeaks of the aforementioned components are located at 854.0 and 856.3 eV, being highly characteristic for nickel(II) oxide NiO and nickel(II) hydroxide Ni(OH)_2_, respectively [38,39,40,41]. This is further confirmed by the location of the Ni satellite peak at 861.3 eV. The small signal intensity originates from substantial surface coverage with adventitious carbon layer, including oxidized carbon species. At the same time, the ion gun etching was not performed, due to significant observable changes in surface chemistry caused by the Ar^+^ ions used for sputtering purposes. These findings are further confirmed by the O1s spectra visible at Figure 9c, where the O_(B)_ peak was ascribed to the nickel hydroxide species, while the dominant O_(C)_ component represents the organic carbon-oxygen bonds and carbonates formed during air exposure. The detailed analysis is summarized in Table 3.

The one weeklong exposure of the analyzed sample in 0.05 M NaCl solution resulted in partial modification of the observed sample surface chemistry. While the chemical state of Ni and O peaks appears to be unchanged, the photoelectron intensity count is significantly improved, suggesting partial removal of the carbon-based corrosion products layer. Nickel appears primarily in the form of Ni(OH)_2_ (symbol Ni_(C)_), with the NiO to Ni(OH)_2_ ratio slightly decreasing from 1:2 to 1:3. Furthermore, the third component Ni_(A)_ emerges as a result of the exposure, its location is characteristic for metallic nickel. Again, the shape of O1s spectra corroborates the results, revealing an increased intensity of hydroxide species O_(B)_ but also nickel oxide species O_(A)_ at approximately 529.5 eV. The amount of adventitious carbon dropped twice.

A characteristic feature of XPS measurements carried out after blank Ni coating exposure to corrosive media is the appearance of the Cu substrate beneath the layer (see Figure 9b). Two peak doublets could be deconvoluted on the basis of the spectra shape: Cu_(A)_ with Cu2p_3/2_ peak at 932.5 eV and Cu_(B)_ at 933.5 eV. The position of the Cu_(A)_ component is characteristic both for metallic copper as well as for Cu_2_O [42], on the other hand, the Cu_(B)_ component was ascribed to CuO oxides according to previous literature findings [43,44,45,46]. The total copper contribution reached 3.5 at.%.

Similar to blank Ni coating, detailed high-resolution XPS analysis was also carried out for the composite Ni coating, where the analysis was expanded with the Ce3d and Mo3d spectral range. The results of the aforementioned analysis are presented in Figure 10 with the spectral deconvolution summarized in Table 4.

Unlike the blank Ni coating, the nickel chemistry on the surface of the composite coating prior to exposure in corrosive media is well developed, with the total Ni contribution ranging from 38 at.% compared to merely 5 at.% observed in the absence of microflake functionalization. Furthermore, the metallic Ni peak Ni_(A)_ is also well developed, proving significantly higher corrosion resistance of the protective layer under atmospheric conditions. Another difference is the strong contribution of the Ni_(B)_ component, revealing increased NiO to Ni(OH)_2_ ratio of 1.2:1 (compared to 1:2 for blank Ni prior the exposure).

As mentioned previously, the galvanic Ni coating was modified using a cerium molybdenum oxide hydrate compound. The chemistry resulting from Ce and Mo surface modification is shown in Figure 10c,d, respectively. Comparison with the precursor compound analysis denoted as the reference confirms high spectral similarity, proving functionalization, did not introduce significant chemical changes. The cerium Ce3d spectra reveals high complexity, the deconvolution was performed using two peak doublets, Ce_(A)_ and Ce_(B)_. Both of these were ascribed to the Ce^3+^ component on the basis of the value of peak BE (881.8 and 885.7 eV) and similar intensity, nearly at a 1:1 ratio [47,48,49]. It is possible, however, that composite coating before exposure contains some Ce^4+^, which can be inferred from the lowered position of the component at the energy 886 eV.

A similar comparison was made in the Mo3d spectral range for cerium molybdenum oxide hydrate compound and for Ni composite coating functionalized with these microflakes. Both analyses reveal the dominant presence of the component ascribed in Figure 10d as Mo_(B)_. The peak location at 232.7 eV is a very good match with the literature BE value of Mo^6+^, typically reported in MoO_3_ oxides and (MoO_4_)_3_^2−^ molybdates [50,51,52]. The second component is common for both the analyzed samples, Mo_(A)_, is negatively shifted at approximately −2.1 eV, being characteristic for the Mo^5+^ component. The Mo^6+^:Mo^5+^ ratio of 2.8:1 remains nearly intact. Finally, the surface analysis of Ni composite coating revealed the appearance of Mo_(C)_ component at 229.5 eV bound with Mo^4+^ oxides. Its absence in cerium molybdenum oxide hydrate microflakes powder may indicate partial reduction of molybdenum during the electrodeposition process [53,54,55]. According to the literature, the reduction of molybdenum could take place during electrodeposition, because the shape of the voltammetric curve and currents achieved differed from those recorded in a blank Ni electrolyte (Figure 3).

The Ni chemistry on the surface of the composite Ni coating changed as a result of the exposure to the 0.05 M NaCl solution. The second analysis carried out after a one-week period revealed a double diminished Ni contribution associated primarily with the observable removal of the Ni_(B)_ component from the coating surface. It is possible that NiO got, in part, reduced to the metallic Ni in local cathodic areas and, in part, transformed to Ni(OH)_2_. At the same time, the amount of molybdenum oxides within the protective coating nearly completely disappeared as a result of the corrosion process. The removal of cerium molybdenum oxide hydrate microflakes was further confirmed by significant removal of cerium species, however, it appears that the corrosion process is less selective towards it, since the spectra was still recognizable after the exposure, with nearly intact chemistry. Finally, the secondary effect of the corrosion process is the appearance of the Cu oxides, Cu_(A)_ and Cu_(B)_. Not only the total contribution of Ni, but also Cu, is significantly increased in comparison to blank Ni coating, indicating smaller thickness of the protective layer and/or locally reduced coverage resulting from consumption of cerium/molybdenum species during the corrosion process. 

### 3.4. Chemical Analysis of Corrosive Solutions

To verify the hypothesis regarding the selectivity of Ce and Mo leaching from microflakes embedded in the composite coating, an additional experiment was planned consisting of a comparative analysis of the composition of NaCl solutions both before and after the corrosion process. For this purpose, blank Ni and Ni composite coatings were immersed for 7 days in 0.05 M NaCl solutions (100 ml volume for each sample). One series of samples was just immersed and left for 7 days, a second one was immersed with an additional continuous stirring of the solution. After 7 days, all solutions were filtered and analyzed by ICP-OES and ICP-MS. The focus was on the comparative analysis of: Ni, Cu, Ce, and Mo content, with blank 0.05 M NaCl as the reference. The results are summarized in Table 5. 

The results presented in Table 5 clearly indicate that Ce did not go into the solution in any significant amounts (similar level before and after corrosion process). At the same time Mo content increased from ca. 5 to 370–390 µg/L. This analysis shows that molybdenum is not only leached from microflakes, but also significant amounts of Mo passed into the corrosive solution. This behavior is quite possible, because the SEM observation of composite coating after 168 hours exposure clearly indicates that some of the microflakes have probably been leached away and those that were visible have been partially dissolved, leaving something like a skeleton/frame (Figure 11a). Furthermore, XPS analysis seems to confirm that molybdenum, when released from micro-flakes, does not form a stable oxide layer on the surface of the coating. Such selective dissolution probably caused voids (well visible in Figure 11a) in the coating, which resulted in a lower corrosion resistance of this composite material.

The results related to Ni concentration (Table 5) very likely to confirm that Ni almost completely forms the layer of corrosion products. Only point corrosion products were visible on the surface of blank Ni coating (Figure 11b). Its (nickel) presence in the solution after corrosion at a level similar to that determined in the blank 0.05 M NaCl solution only confirms this assumption. Different behavior was observed for copper (originating from the substrate). In the case of blank Ni coating, Cu was not observed to pass into the NaCl solution, which, in the context of the results of the XPS surface analysis, only confirmed that this element is part of the relatively stable corrosion products on the surface of this coating. Only in the case of composite coating, copper was determined at the level of 14–29 µg/L, which means that this element partially passed in the form of an ion into the solution. This is even more possible because during the dissolution of the microflakes, the copper substrate was gradually exposed to chloride ions.

## 4. Conclusions

In summary, it can be concluded that:(1)DES-based plating baths are an excellent environment for the electrodeposition of metal composite coatings.(2)The addition of cerium molybdenum oxide hydrate microflakes to a plating bath modifies the cathodic process sufficiently to obtain nanocrystalline nickel composite coatings with a smaller (6.3 nm) crystallite size than blank Ni coating (10.4 nm).(3)The codeposition of large CeMo oxide microflakes caused microcracks in the coating and some deterioration of the protective properties of composite coating.(4)Charge transfer resistance for both types of coatings increased over 7 days exposure in 0.05 M NaCl solution. However, even after this time, the *R*_ct_ for a blank Ni exceeded five times (581 kΩ cm^2^) that of a composite one (103 kΩ cm^2^).(5)On the basis of spectroscopic studies (XPS, ICP-OES and ICP-MS), it can be stated for composite Ni coating that in the course of corrosion, NiO transforms to Ni(OH)_2_, while molybdenum oxides nearly completely disappear from the surface of composite coating. The removal of cerium species was less than that of the molybdenum species, which suggests that the corrosion process is more selective towards Mo. The smaller thickness of the protective layer and locally reduced coverage results from the consumption of the Ce/Mo species during the corrosion process.(6)It would be interesting to develop the described research towards the synthesis of nanoflakes (or nanowires) of mixed cerium and molybdenum compounds. Reducing the particle size to the nano-scale would result in a composite coating with a better dispersed oxide phase. This, in turn, would improve the tightness of the metal coating.

## Figures and Tables

**Figure 1 materials-13-00924-f001:**
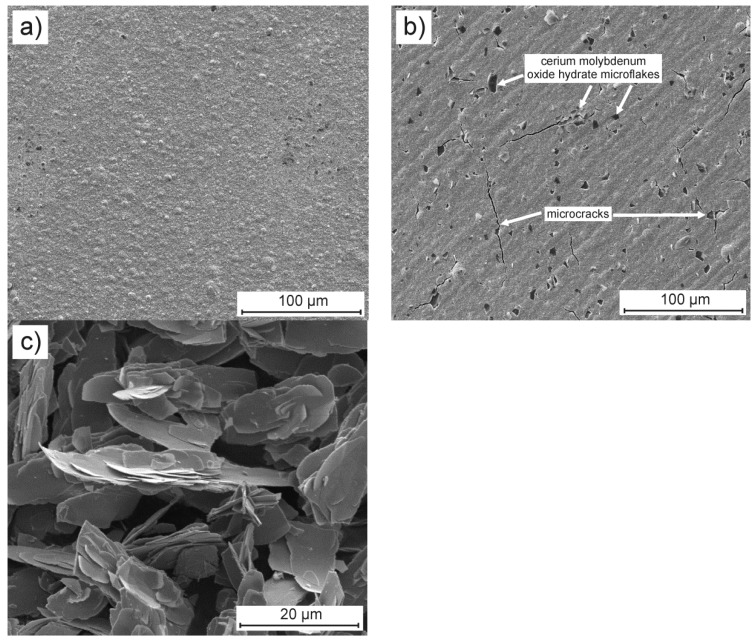
Surface morphology of (**a**) blank and (**b**) composite Ni coatings deposited on the copper substrate at a current density of 6 mA cm^−2^ at 70 °C for 1 h, and (**c**) as-synthesized cerium molybdenum oxide hydrate microflakes for comparison. Pictures were recorded in a SE mode of SEM.

**Figure 2 materials-13-00924-f002:**
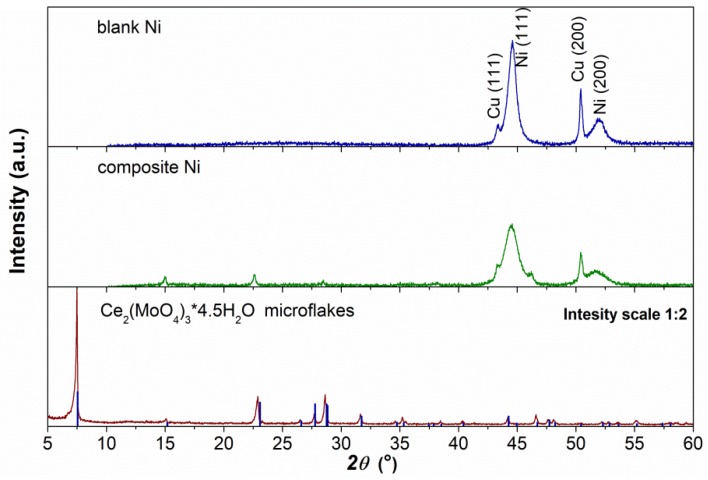
X-ray diffractograms for blank and composite Ni coatings deposited on the copper substrate at a current density of 6 mA cm^−2^ at 70 °C for 1 h. A powder diffraction pattern for as-synthesized cerium molybdenum oxide hydrate is also shown at the bottom of the figure for comparison.

**Figure 3 materials-13-00924-f003:**
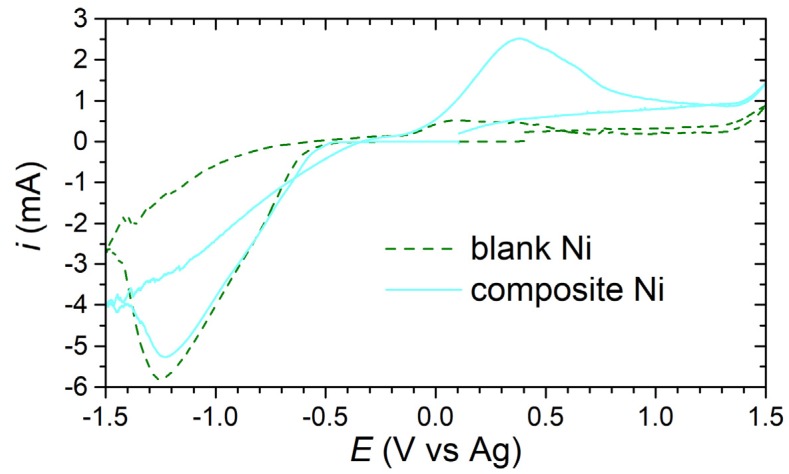
Cyclic voltammetry curves recorded in a blank Ni plating bath and suspension bath containing 2 g dm^−3^ Ce_2_(MoO_4_)_3_ 4.5H_2_O microflakes, at 70 °C on a rotating disk electrode with Pt tip (3 mm diameter), at 600 rpm.

**Figure 4 materials-13-00924-f004:**
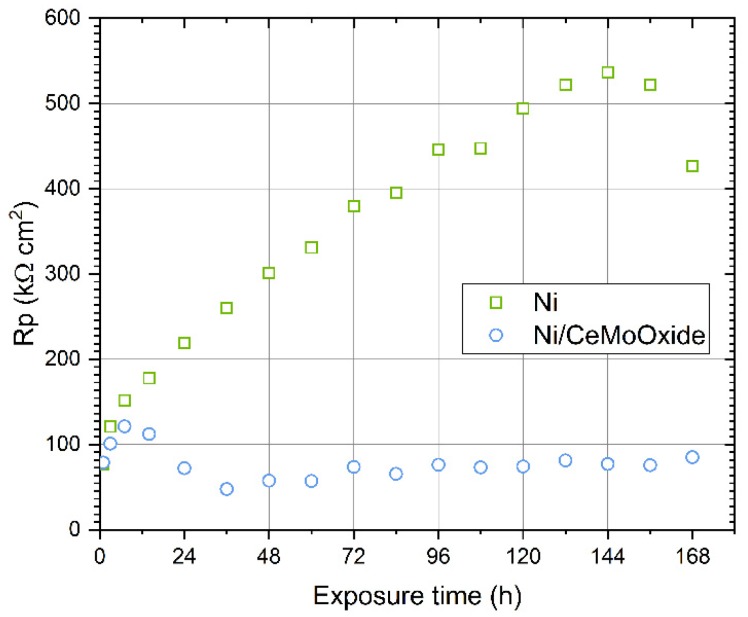
Polarization resistance (*R*_p_) recorded for blank and composite Ni coatings during 168-hour exposure in 0.05 M NaCl solution.

**Figure 5 materials-13-00924-f005:**
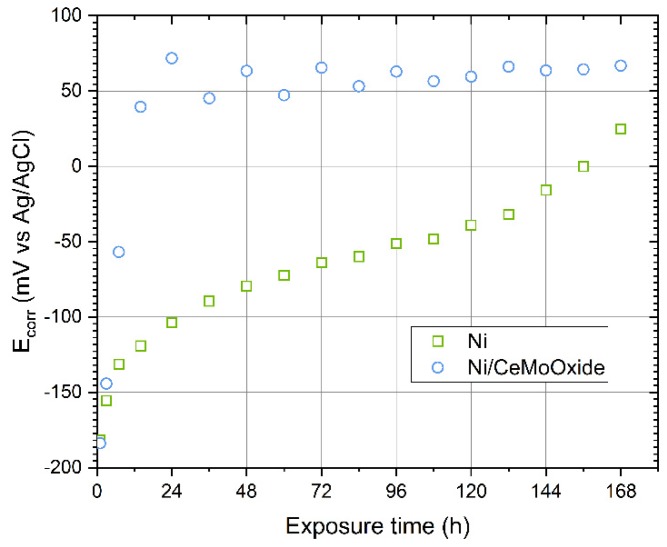
Corrosion potential (*E*_corr_) recorded for blank and composite Ni coatings during 168-hour exposure to 0.05 M NaCl solution.

**Figure 6 materials-13-00924-f006:**
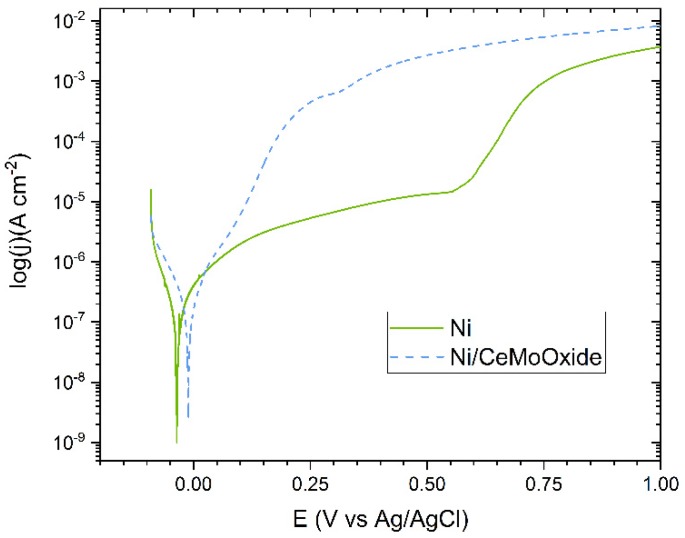
Potentiodynamic polarization curves recorded for blank and composite Ni coatings after 168-hour exposure to 0.05 M NaCl solution.

**Figure 7 materials-13-00924-f007:**
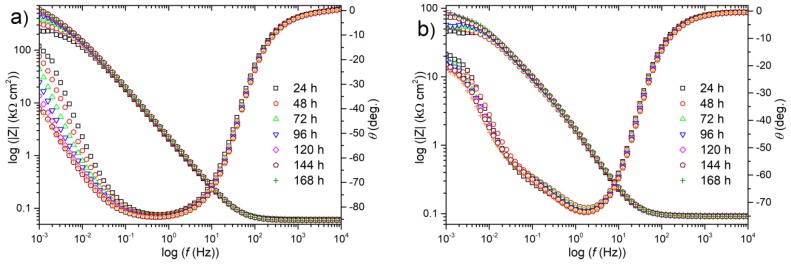
Bode representation of the impedance spectra recorded for (**a**) blank and (**b**) composite Ni coatings for 168 hours exposure in 0.05 M NaCl solution.

**Figure 8 materials-13-00924-f008:**
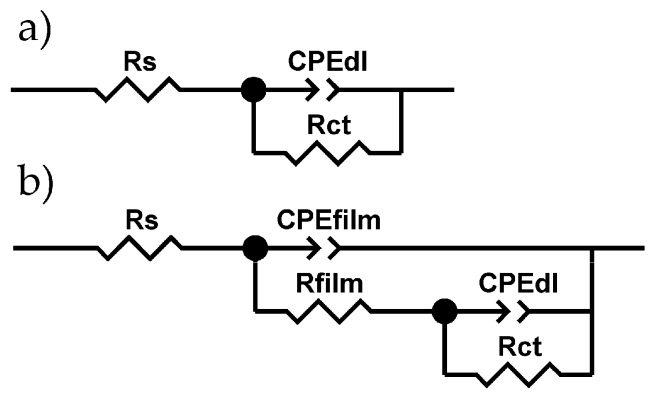
Electric equivalent circuits used for fitting the experimental spectra of (**a**) blank and (**b**) composite Ni coatings.

**Figure 9 materials-13-00924-f009:**
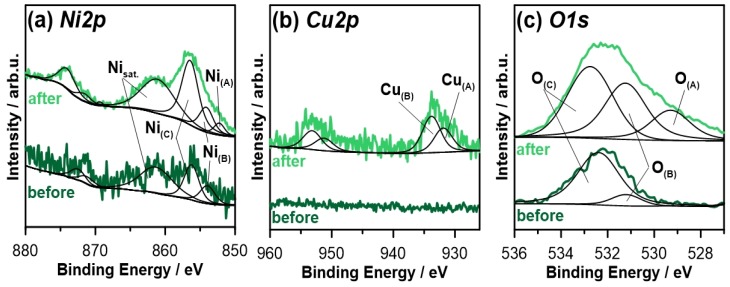
High-resolution XPS spectra recorded for blank Ni coating in (**a**) Ni2, (**b**) Cu2p, and (**c**) O1s peak binding energy range, before and after the exposure to 0.05 M NaCl.

**Figure 10 materials-13-00924-f010:**
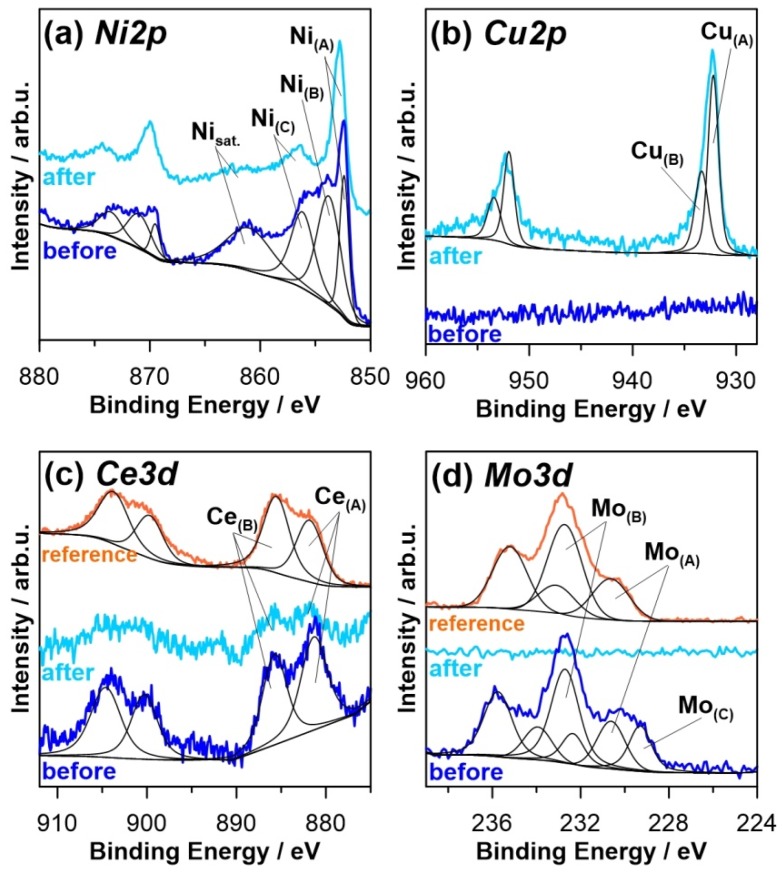
High-resolution XPS spectra recorded for composite Ni coating in (**a**) Ni2p, (**b**) Cu2p, (**c**) Ce3d, and (**d**) Mo3d peak binding energy range, before and after the exposure to 0.05 M NaCl. The chemistry of synthesized cerium molybdenum oxide hydrate powder used for electrodeposition is also shown in (**c**) Ce3d and (**d**) Mo3d spectra (“reference” spectra).

**Figure 11 materials-13-00924-f011:**
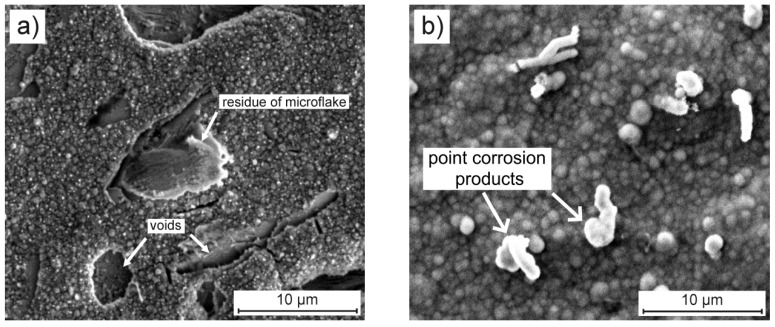
Surface morphology of (**a**) composite and (**b**) blank Ni coatings (deposited on the copper substrate at a current density of 6 mA cm^−2^ at 70 °C for 1 h) after one-week exposure to 0.05 M NaCl solution.

**Table 1 materials-13-00924-t001:** Values of electric elements calculated for EEC of blank Ni coating.

Time (h)	*R*_s_/ Ω cm^2^	CPE_dl, Y0_/ Ω^−1^ cm^−2^ s^α^	CPE_dl, α_	*R*_ct_/ kΩ cm^2^
24	61	9.0 × 10^−5^	0.94	214.8
48	61	8.6 × 10^−5^	0.94	299.6
72	61	8.4 × 10^−5^	0.93	394.8
96	60	8.3 × 10^−5^	0.93	478.8
120	59	8.2 × 10^−5^	0.93	552.4
144	59	8.1 × 10^−5^	0.94	604.8
168	58	8.2 × 10^−5^	0.93	580.8

**Table 2 materials-13-00924-t002:** Values of electric elements calculated for EEC of composite Ni coating.

Time (h)	*R*_s_/ Ω cm^2^	CPE_film, Y0_/ Ω^−1^ cm^−2^ s^α^	CPE_film, α_	*R*_film_/ kΩ cm^2^	CPE_dl, Y0_/ Ω^−1^ cm^−2^ s^α^	CPE_dl, α_	*R*_ct_/ kΩ cm^2^
24	92	1.0 × 10^−4^	0.95	5.6	8.5 × 10^−5^	0.66	74.8
48	92	1.0 × 10^−4^	0.94	5.9	9.9 × 10^−5^	0.69	59.4
72	93	9.9 × 10^−5^	0.94	7.0	9.8 × 10^−5^	0.68	77.1
96	92	9.8 × 10^−5^	0.94	7.9	9.2 × 10^−5^	0.67	81.9
120	92	9.7 × 10^−5^	0.94	8.6	8.6 × 10^−5^	0.67	84.6
144	91	9.6 × 10^−5^	0.94	8.2	8.4 × 10^−5^	0.65	92.2
168	91	9.5 × 10^−5^	0.94	8.8	7.9 × 10^−5^	0.64	102.7

**Table 3 materials-13-00924-t003:** Spectral deconvolution performed for blank Ni coating.

Element		BE / eV	Blank Ni Before	Blank Ni After
Ni2p	Ni_(A)_	852.4	-	1.2
	Ni_(B)_	854.0	1.9	3.2
	Ni_(C)_	856.3	3.7	9.7
Cu2p	Cu_(A)_	932.5	-	1.4
	Cu_(B)_	933.5	-	2.1
O1s	O_(A)_	529.5	-	15.6
	O_(B)_	530.8	12.2	30.5
	O_(C)_	532.4	82.2	36.7

**Table 4 materials-13-00924-t004:** Spectral deconvolution performed for composite Ni coating.

Element		BE/eV	NiCeMo Before	NiCeMo After	CeMo Reference
Ni2p	Ni_(A)_	852.5	11.2	5.8	-
	Ni_(B)_	853.7	14.4	3.8	-
	Ni_(C)_	856.3	12.2	5.1	-
Cu2p	Cu_(A)_	932.5	-	5.1	-
	Cu_(B)_	933.3	-	2.9	-
Ce3d	Ce_(A/B)_	881.8/885.7	6.1	2.4	11.6
Mo3d	Mo_(A)_	230.6	1.2	-	5.4
	Mo_(B)_	232.7	3.2	-	15.1
	Mo_(C)_	229.3	1.2	-	-
O1s	O_(A)_	529.5	21.8	0.3	6.0
	O_(B)_	530.8	21.3	11.6	22.5
	O_(C)_	532.4	7.4	63.0	39.4

**Table 5 materials-13-00924-t005:** Concentrations of Ni, Cu, Ce, and Mo in a corrosive solution.

Element	Units	Blank (Demi Water + 0.05M NaCl)	Blank Ni	Blank Ni_m *	Composite Ni	Composite Ni _m *
Ni	µg/L	<2.0 ^2^	2620 ± 370 ^1^	4230 ± 590 ^1^	2730 ± 380 ^1^	6000 ± 840 ^1^
Cu	µg/L	2.4 ± 0,4 ^2^	2.1 ± 0.3 ^2^	2.2 ± 0.3 ^2^	29 ± 4 ^2^	14 ± 2 ^2^
Ce	µg/L	<0.1 ^2^	<0.1 ^2^	< 0.1 ^2^	0.33 ± 0.07 ^2^	0.13 ± 0.03 ^2^
Mo	µg/L	5.5 ± 0.8 ^2^	5.3 ± 0.8 ^2^	17 ± 3 ^2^	390 ± 55 ^2^	374 ± 52 ^2^

^1^ ICP-OES analysis. ^2^ ICP-MS analysis. * Solution was mechanically stirred for 7 days exposure of the coatings.

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
