# Peer review of "Ni/cerium Molybdenum Oxide Hydrate Microflakes Composite Coatings Electrodeposited from Choline Chloride: Ethylene Glycol Deep Eutectic Solvent"

_materials, 2020, doi:10.3390/ma13040924_

Round 1

Reviewer 1 Report

The structure of the scientific report is good and well-understood. The aim is clarified. The introduction summarizes well.

Reviewer 2 Report

The submitted manuscript entitled ‘Ni/cerium molybdenum oxide hydrate microflakes composite coatings electrodeposited from choline chloride: ethylene glycol deep eutectic solvent’ deals with the production and investigations (mainly corrosive investigations) of a composite coating. The manuscript is interesting, during its review only a few technicalities arose.

- Please always let a space between the value and its unit, except in the case of ‘°C’ and ‘%’.

- How were the experimental parameters determined? By trials or by theory?

- The Materials and Methods section is quite long, please shorten.

- Please identify the phases in fig 1b. The cracks in fig 1b are troublesome. The final materials should contain no cracks.

- Labels in fig 2 are too small to read.

- Please identify the phases in fig 11.

Reviewer 3 Report

To Editor and Authors,

In review, I received the manuscript entitled “Ni/cerium molybdenum oxide hydrate microflakes composite coatings electrodeposited from choline chloride: ethylene glycol deep eutectic solvent” considered for publication in MDPI Journal “Materials”. The article is well-written, and it is worth publishing. Nevertheless, there are few concerns that have to be resolved before publication, namely:

Some sentences are a bit crude and need rewriting to avoid ambiguity, for example:

P2/L49 “which are completely various than these used in aqueous plating baths” – probably different?

P2/L69 “It causes receipt of product high pure not include cerium molybdate oxides”

The introduction is nicely covering in detail the current state and the background of the topic, although I would suggest avoiding starting the paragraphs with “interesting subject of research” or similar. With a slight rewrite, this part of the text can perform much more professional, but it is just a suggestion.

Some other comments are:

P2/L80: potential self-healing properties should be further explained (why you believe that) or properly referred to previous research (reference).

L117: “XRD patterns” as XRD is spectroscopy method, you are recording spectra, not patterns.

P3/L103, L112: “in temperature” => at

L123: X-Ray is usually written X-ray

L165: “was characterized by quite homogeneous structure” needs rewriting. Probably “as”? Quite homogenous is too undetermined, please be more concise, you can just skip this part and adjoin with the next sentence. Please also check the commas through the text, like here: “spherical particles, which, were noticeably…”.

L173: besides Fig 1 subtitle as “photomicrograph”, please include the SEM mode used for recording these micrographs (SEI, BEI, …). The statement “It can be 173 seen that these are not perfectly formed and loose flakes” needs to be more concise: I guess you are stating that flakes are loose, but the sentence does not read like that in the current form.

L183: “crystal planes visible at 2θ about 44.6°, 51.9°.” “About” is here unnecessary, I think, as the position of diffraction peaks are very precisely determined in the literature, and apparently it is a good match with your results.

Regarding Fig 2 I suggest additional text at the XRD method, stating in which range the spectra were collected. From the Fig 2 I assume the Ni-phases were recorded from 10 – 60 °, while for Ce composite from 5 – 60°. Just to explain why the first sharp peak at composite is missing at about 7 °. 

L241, L253: The statement: “It is possible that such a direction of changes resulted from qualitative changes and changes in the morphology of the passive layer / layer of nickel oxidation products” can be nicely supported by the observation of possible morphology change by SEM of surface or cross-section. Although I understand the difficulties related to additional analyses, please reconsider the possibilities of investigating the surfaces after the corrosion test (if samples are still available).

L273: does the “damaged coating” refers to the cracks observed on as-deposited surfaces? This would probably be worth to mention here.

Fig9: I suggest using a different colour for fitting curves, black would be OK. It is maybe overkill, but now some of the fitting curves are overlapping with the spectra, and it’s hard to guess where it is (like in b around 935eV). But I don’t have any scientific objections regarding your interpretation of the XPS results… 

L326: Ar+ ions are probably not reducing the Ni-oxides, or are they? I found especially the layered Ni-oxide polymorph highly sensitive to any electron or ion bombardment, but this usually just affects their structure and not their chemistry. Forgive me if I’m wrong here, but would not be possible that during the etching the oxide layer is being either selectively etched, maybe even contaminated (but not reduced)? Overall, maybe just a rephrasing is needed here.

L340: What is your explanation of carbon source, and the mechanism of its removal by NaCl solution?

L414: “microscopic observation of SEM composite” => probably SEM observations of composite? Similarly, rewrite L416.

L418: is “leaks” proper terminology? I would use here spallation, referring to scales? Of course, it strongly depends on the mechanism that produces such features (pure corrosion or delamination).

Figure 11a seems to not be referred to in the text.

L444: “exceeded fife times” => five

Overall, with some adjustment of the text, I find the paper interesting for publishing due to their presentation of the different passivation layer formation in combination with additives/composites. Although the protective coating can be protective only when a continuous film is formed, the introduction of any particles will cause spallation. Therefore, although the results were quite expected, the research is confirming the assumpted hypothesis, and it is contributing to scientific knowledge. 

Regards, Reviewer

Reviewer 4 Report

This paper present Ni/cerium molybdenum oxide hydrate microflakes composite coatings electrodeposited from choline chloride: ethylene glycol deep eutectic solvent. I would recommend this paper to be published as the suggestions and some additional info been added as listed below:

The geometry of the specimen should be provided with a specimen sketch. The methodology part should be summarized by providing a flowchart. Label Figure 1 to emphasize the crack location and how could be solved. Explain how this solution might affect the fatigue corrosion. I would suggest to cite several advanced techniques in modelling crack propagations based on energy released rate in composite structures such as using XFEM for your recommendation. Abdullah, Nur Azam and Akbar, Mahesa and Wirawan, Nanda and Curiel-Sosa, Jose Luis (2019) Structural integrity assessment on cracked composites interaction with aeroelastic constraint by means of XFEM. Composite Structures. ISSN 0263-8223 Abdullah, Nur Azam and Curiel-Sosa, Jose Luis and Taylor, Zeike A. and Tafazzolimoghaddam, Behrooz and Martinez Vicente, J.L. and Zhang, Chao (2017) Transversal crack and delamination of laminates using XFEM. Composite Structures, 173. pp. 78-85. ISSN 0263-8223 MC Serna Moreno, JL Curiel-Sosa, J Navarro-Zafra, JL Martínez Vicente, JJ López Cela (2015) Crack propagation in a chopped glass-reinforced composite under biaxial testing by means of XFEM Composite Structures.
